# Review on Functional Testing Scenario Library Generation for Connected and Automated Vehicles

**DOI:** 10.3390/s22207735

**Published:** 2022-10-12

**Authors:** Yu Zhu, Jian Wang, Fanqiang Meng, Tongtao Liu

**Affiliations:** College of Computer Science and Technology, Jilin University, Changchun 130012, China

**Keywords:** connected and automated vehicles, simulation test, test scenario generation, road generation, dynamic scenario generation

## Abstract

The advancement of autonomous driving technology has had a significant impact on both transportation networks and people’s lives. Connected and automated vehicles as well as the surrounding driving environment are increasingly exchanging information. The traditional open road test or closed field test, which has large costs, lengthy durations, and few diverse test scenarios, cannot satisfy the autonomous driving system’s need for reliable and safe testing. Functional testing is the emphasis of the test since features such as frontal collision and traffic sign warning influence driving safety. As a result, simulation testing will undoubtedly emerge as a new technique for unmanned vehicle testing. A crucial aspect of simulation testing is the creation of test scenarios. With an emphasis on the map generating method and the dynamic scenario production method in the test scenarios, this article explains many scenarios and scenario construction techniques utilized in the process of self-driving car testing. A thorough analysis of the state of relevant research is conducted, and approaches for creating common scenarios as well as brand-new methods based on machine learning are emphasized.

## 1. Introduction

Connected and automated vehicle (CAV) specifically refers to a new generation of vehicles equipped with on board unit (OBU), LiDAR and other sensors, controllers, actuators, and other devices to achieve information exchange and share between vehicles and people, roads, and backstage, which can eventually replace human operation, as shown in Figure 1. With the development of perception technology [1,2,3], decision control technology [4,5,6], and vehicle to everything (V2X) [7,8,9] communication technology, the impact of CAV [10,11,12,13] in people’s daily lives is becoming more and more widespread.

On the one hand, CAV uses sensors to sense the presence of other traffic participants in the surrounding environment; the sensors mainly include cameras, LiDAR, millimeter wave radar, etc. Since a single sensor is easily affected by the physical environment, for example, in rain, snow, and haze, LiDAR penetration is poor at providing accurate sensing results, so it needs to rely on millimeter wave radar and camera for the main sensing task. Furthermore, sensing fusion technology can enable CAV to obtain more accurate environmental information, but the sensing capability still has some limitations in extreme bad weather conditions.

On the other hand, CAV can also send and receive messages through the direct communication interface (PC5) of OBU and cellular network communication interface (Uu) for the purpose of acquiring external information and sending the self-vehicle status outward. Current V2X communication mainly includes vehicle to vehicle (V2V), vehicle to infrastructure (V2I), etc. V2V means vehicles communicate with each other directly through OBU. V2I means that a roadside unit (RSU) obtains information from roadside sensors, traffic lights, road signs, etc. through a local area network (LAN) and broadcasts it. CAV can communicate with RSU to obtain more road information, emergency vehicle (EV) information, and vulnerable road user (VRU) information. At the same time, an intelligent transportation system (ITS) can access the city traffic management center and Cellular-V2X (C-V2X) can give some related instructions through the base station.

CAVs can obtain information about traffic participants in the surrounding environment through the above-mentioned means. Multiple CAVs combined with their applications, such as trajectory planning, autonomous driving functions, and hazard warnings, can create a more efficient and safe traffic environment. However, current autonomous driving algorithms are not mature, and early warning algorithms (e.g., forward collision warning) are not well developed. Instead of improving the traffic driving environment, imperfect driving algorithms may cause more serious traffic accidents. The automatic driving function test mainly includes overtaking, merging, following driving, etc. The functional test contents of V2X are mainly shown in Table 1, where P denotes pedestrian and X denotes everything.

Safety is a prerequisite for the mass market launch of CAVs. Due to the sparseness of safety accidents, real-world testing is costly, inefficient, and requires testing of high-risk or accident scenarios. However, self-driving cars need to accumulate hundreds of millions or even hundreds of billions of kilometers of test miles to effectively verify their safety performance [14,15]. Therefore, it is not realistic to rely solely on real-world testing to gradually improve driving algorithms. Simulation testing has been widely adopted to speed up the testing process. Simulation testing [16,17,18] greatly reduces the economic and time costs of testing and increases the safety of the testing process. In the actual testing, we have encountered the problem of inaccurate/untimely GPS positioning. When testing the warning function, we found that some of the tested parts (OBU) did not consider the elevation of the vehicle, and when the vehicle was located near an overpass, the position resolution of the vehicle would be wrong (spatial misalignment), leading to false triggering of the warning.

The difficulty of simulation testing is building a reasonable test case library. According to test types, test scenarios can be divided into functional scenarios, accident scenarios, and regulatory scenarios. Among them, accident scenarios come from the reproduction of real traffic accident data. The regulatory scenario develops corresponding tests according to local laws and regulations with clear safety boundaries. The functional scenario is the test for different functions of CAV, which is the main part of the test and is also the main scenario discussed in this paper.

At present, the method of manually building test scenarios is still mainly used in simulation testing. The manual construction of test scenarios can precisely combine the test requirements. However, the efficiency of manual scenario building is low. If you want to cover a test function completely, it may take a lot of time. There are many researchers exploring how to achieve the automatic generation of test scenarios. Meanwhile, there are more related generation methods; however, so far, there is no work to organize, compare, and summarize the existing scenario generation methods in a systematic way.

Firstly we make a clear definition and terminology related to the existing scenario generation. Secondly, we introduce the methods of road generation, including (1) actual field collection, (2) road extraction from remote sensing imagery, and (3) road extraction from OpenStreetMap files. Thirdly, we introduce the methods of dynamic scenario generation, including (1) combinatorial testing, (2) knowledge-based generation, (3) driving behavior-based generation, and (4) data-driven generation. The corresponding examples are also given to illustrate some of the mainstream methods.

## 2. Scenario Definition

In the subject of autonomous driving, scenario-based autonomous driving simulation testing has been seen as particularly promising. Menzel et al. [19] proposed a three-level hierarchy of “functional-logical-concrete” for scenarios, as shown in Figure 2. Taking the test scenario of a vehicle driving in a straight line as an example, the functional scenario is “vehicle driving straight”, the logical scenario is “vehicle driving straight, speed range is [0, 120] km/h”, the concrete scenario is “vehicle driving straight at 10 km/h”.

A functional scenario for an automated driving simulation test project is produced by an abstract language description. In the conceptual stage, functional scenarios may be utilized to define an autonomous driving simulation test project and analyze project risks and hazards. Using linguistic scenario notation, functional scenarios define the domain’s entities and their connections. A consistent vocabulary with names for the various things and the connections between these items must be defined due to the utilization of linguistic descriptions. Before testing, the tester needs to confirm the list of test functions and make a list of functional scenarios. One of the more critical features examined in the automatic driving functional test is Autonomous Emergency Braking (AEB). The AEB functioning scenario is as follows: when the host vehicle (HV) senses a potential accident, the car immediately brakes, assuring the driver’s safety. When the HV is moving in a straight line and the safety distance from the remote vehicle (RV) is short, the AEB feature is activated, as shown in Figure 3.

In the context of automated driving simulation testing, functional scenarios described in an abstract language need to be represented in state space and need to be converted to the corresponding data format of the simulation environment to generate logical scenarios. The canonical logic scenes must be described in the formal notation of the state space. Furthermore, parameters must be defined by ranges of values, which can be better described by specifying the relationship between probability distribution and parameter range for each parameter. Different assisted driving functions have different speed intervals, and the assisted function will not turn on when the speed is too low or too high. Taking the Autonomous Emergency Braking (AEB) as an example, the recommended operating range of AEB in urban areas is [0, 70] km/h [20]. The definition of parameter ranges for logical scenarios can clarify the scope of the test, avoid meaningless testing, and improve the efficiency of the test.

The parameters of the logical scenarios are specified and the logical scenarios are converted into concrete scenarios for simulation testing. The concrete scenarios explicitly describe the functional scenarios at the state space level, and the relationships between traffic participants and related entities can be represented with the help of specific values of each parameter in the state space. Again, using AEB as an example, its concrete scenario will specifically describe the speed of the vehicles, the spacing between the vehicles, etc. For example, the host vehicle speed in Figure 3 is 70 km/h and the spacing between the two vehicles is 30 m at 24.150 s. Another concrete scenario is intersection collision warning, as shown in Figure 4. The host vehicle is passing through the intersection at 53 km/h, and a black truck (20 km/h) suddenly rushes out from the left side of the intersection at 17.015 s. This scenario is also extremely dangerous and is used to test the warning capability of the vehicle in extreme working conditions. It is difficult to test such hazardous scenarios in actual road tests, reflecting the importance of simulation testing.

In this article, we will refer to scenario files that can be run directly in the simulator as a formatted scenario.

To avoid overlooking many important test scenarios, we systematically review issues based on widely accepted test models (e.g., the V-model) and focus on requirements (for each development phase), examining the methods used to generate scenarios that satisfy those requirements, as shown in Figure 5. Since the testing purposes of different scenarios are different, here is a cut-in scenario as an example. In the functional scenario, it is mainly necessary to specify all the sub-functions, use cases, system architecture, and operational design domain (ODD) under the scenario. The sub-functions of the cut-in function include the vertical control and horizontal control of the vehicle. Test cases can be divided into emergency cut-in, mandatory cut-in, and lane change cut-in according to urgency. The test cases can be divided into close cut-in and long cut-in according to the cut-in distance. The system architecture of the cut-in function includes sensors, controllers, and actuators. The ODD of the cut-in function includes (1) highway or urban expressway, (2) high precision map is valid, and (3) clear lane lines, no road traffic hazard signs, no signal lights, and normal weather.

HARA refers to hazard analysis and risk assessment. Hazard Analysis is the first step of the process and is used to assess the Automotive Safety Integrity Level (ASIL) level of the risk. The purpose of hazard analysis is to determine the ASIL level and the required safety status. Risk Assessment contains two aspects: (1) hazard identification and (2) risk factors that may cause hazards. We analyze the cui-in function to obtain the vehicle-level hazards, as shown in Figure 6.

Fault Tree Analysis (FTA) is one of the most widely used methodologies for determining a system’s dependability. It is a design analysis and assessment approach for determining various possible combinations of system failure causes and their likelihood of occurrence, as well as calculating system failure probability and taking appropriate actions to enhance system dependability. We analyze the cui-in function to obtain the FTA model. We may deduce that the particular causes could include sensors, sensing algorithms, planning/control algorithms, actuator performance constraints, communication issues, user misbehavior, and so on.

Testing the planning algorithm, for example, requires excluding other interfering factors to ensure that other components work properly. For example, to test the impact of communication latency on functionality, we can manually inject network latency and test the impact on performance. Next, analyze all the components that affect the cut-in function and give the expected test range, for example, delay 10–100 ms. Then, obtain all the test intervals and logical scenarios. In the logical scenario set, the parameter intervals are assigned specific values to concrete scenarios. The components and system are tested to verify that the system functions properly. After verifying that the functionality is normal, then the system can be deployed.

Bagschik et al. [21] proposed a five-layer model of the test scenario in conjunction with the VTD simulator. In the five-layer model, the first layer describes the layout of the road, including markers and topological relationships; the second layer contains the traffic infrastructure; and the third layer is a temporary manipulation of the first two layers. The first three layers constitute the description of the map. The process of format carry-over is to convert the elements of layers one to three into OpenDRIVE [22] syntax format. The fourth layer models objects that are not necessarily part of the transportation infrastructure, such as cars and people; the fifth layer describes environmental influences such as weather. The elements of layers four and five are converted to OpenSCENARIO [23] syntactic format.

OpenDRIVE and OpenSCENARIO are the most common scenario file formats at present. The scenario file format supports for the current common vehicle simulation engines are shown in Table 2. The simulator supports these format files directly, or the format file can be formatted through the simulator’s tools. Most simulation software uses the data format of OpenDRIVE for the description of road network structure and OpenSCENARIO for the description of traffic participants and environment. Format conversion refers to the conversion of modeling information from parameter space representation to OpenSCENARIO and OpenDRIVE data formats. CARLA [24] is an open-source simulator for autonomous driving research that also supports OpenDRIVE and OpenSCENARIO formats. CarSim supports the OpenDRIVE format but does not directly support the OpenSCENARIO format. CARLA integrated CarSim support OpenDRIVE and OpenSCENARIO formats. Matlab’s Autonomous Vehicle toolbox (Driving Scenario Designer) also supports OpenDRIVE, but not OpenSCENARIO format.

## 3. Road Generation

In the simulation test of connected and automated vehicles, map files are a relatively easy part to ignore. Map information is playing an increasingly important role in vehicle driving. For example, in collaborative ramp convergence, the vehicle needs to know the information about the ramp entrance; lane departure warning, the vehicle needs to know the distribution of lanes.

The current main approaches for OpenDRIVE map generation include three directions: (1) actual field collection, (2) road extraction from remote sensing imagery, and (3) road extraction from OpenStreetMap files. In this section, these three road generation methods are introduced, and their advantages and challenges are analyzed.

### 3.1. Actual Field Collection

The map collecting equipment includes LiDAR, GPS positioning device, Inertial Measurement Unit (IMU), High Definition (HD) camera, and other sensors. The device also supports Real-Time Kinematic (RTK) technology, which communicates with the base station through 4G technology, and the positioning accuracy can reach the centimeter level, as shown in Figure 7a,b. After determining the acquisition area, the vehicle with the acquisition equipment fixed drives on the road at a uniform speed and collects information about the surrounding environment through the sensors, as shown in Figure 7c. We conducted field collection using the above equipment at a closed test site in Jiading, Shanghai, China. After the acquisition was completed, point cloud data were obtained, as shown in Figure 7d. The collected data was organized, analyzed, and converted to generate HD maps.

To meet the high demand for road spatial data, Y. Shi et al. [25] proposed an automatic road-mapping technique that fuses navigation data, stereo images, and laser scanning data to develop a hybrid inertial survey system (HISS). The multi-sensor-based vehicle-mobile mapping system was demonstrated to be an efficient system in terms of both time cost and road spatial data acquisition results, which can generate high-precision and high-density 3D road spatial data more quickly and at a lower cost. Liang J et al. [26] designed a structured model based on convolutional neural networks that can automate LiDAR and camera data for road extraction, and verified that this method can extract road boundaries with high accuracy and high recall. Homayounfar N et al. [27] developed a hierarchical recurrent network and a new differentiable loss function to solve the problem of extracting road networks from sparse 3D point clouds. Zeybek M et al. [28] achieved automatic extraction of roads using point clouds and machine learning classification methods, and used point clouds to create digital elevation models to extract profile and section elevations, enabling automatic extraction of high-precision road geometric features from unmanned aerial vehicle (UAV) images.

The advantages of the road extraction method for field collection are:1.High accuracy. The positioning accuracy can reach centimeter level, and the lane information is more accurate.2.Highly real-time. Satellite maps and network maps cannot guarantee the real-time update of maps, and there are cases of information errors.3.More information. Information such as lane lines and road information can be captured, which is difficult to extract in remote sensing.

However, the approach to field collection is accompanied by a number of challenges:1.High acquisition requirements. The sensor sensing ability drops abruptly in bad weather conditions.2.Low automation. Collection vehicles or drones require human control.3.High cost. The price of sensors is relatively high, as is the cost of manpower.

### 3.2. Road Extraction from Remote Sensing Imagery

With the development of remote sensing (RS) technology [29,30,31,32], the resolution of remote sensing images increased significantly, making it possible to extract roadways from remote sensing photographs, as shown in Figure 8a. The gathered road information is also becoming more informative with the advancement of machine learning [33] and the creation of open-source datasets [34]. Three-band RGB (red, green, and blue) photos make up the DeepGlobe Road Extraction Challenge dataset [35]. This dataset may be used to train and evaluate the algorithm model for the binary segmentation issue, where the class labels are roads or non-roads.

Classification-based methods usually use geometric, photometric, and textural features of roads. Classification-based methods can be classified as supervised and unsupervised [36].

An approach to supervised learning is Support Vector Machine (SVM). SVM classifiers can extract roads using edge-based characteristics such as gradient, intensity, and edge length; however, numerous studies have noted that the accuracy rate is just a small percentage of the time [37]. SVM techniques are frequently employed for object recognition in RS pictures because of their benefits of structural risk minimization and excellent generalization capacity. The estimate of kernel functions, dimensional space, and training sample selection are some of the challenges associated with applying SVM techniques.

Unsupervised classification techniques offer numerous benefits over supervised ones in terms of addressing classification challenges. Numerous clustering techniques, such as K-means, spectral clustering, mean shift [38], and graph theory [39], are the most often used algorithms. Unsupervised techniques are highly effective yet have poor accuracy since they do not require any prior training or understanding.

Due to the misclassification of roads and other spectrally similar objects such as construction blocks, field blocks, waterways, and parking lots, the classification accuracy is far from good. The classic machine learning methods are becoming more difficult and computationally demanding because of the rising number of buildings in the image, the complexity of the roads, and the sharper shadows of the trees and buildings.

Deep learning and convolutional neural networks have also grown in importance as ways to extract roads from very precise remote sensing images such as machine learning progressed. The DenseUNet model, which has fewer parameters, stronger features, and more accurate extraction results, is increasingly used to extract road networks from remote sensing images [40,41]. Furthermore, its dataset is derived from the screenshots of Google Earth [42], and the automated extraction of roads from Google Earth has become possible. Abdollahi et al. [43] proposed a new deep learning-based automatic network called RoadVecNet, which consists of interconnected UNet networks to perform road segmentation and road vectorization simultaneously. RoadVecNet contains two UNet networks. The first network has a powerful representation capability to obtain more coherent and satisfactory road segmentation maps even in complex urban settings. The second network is connected to the first one and vectorizes the road network by using all the previously generated feature maps. Classification results indicate that the RoadVecNet outperforms the state-of-the-art deep learning-based networks with 92.51% and 93.40% F1 scores for road surface segmentation and 89.24% and 92.41% F1 scores for road vectorization from the aerial and Google Earth road datasets, respectively.

The effectiveness of extracting roads from remote sensing images has substantially increased and is now automated as compared to the field collecting of map data. Despite the fact that lane information is crucial in the smart grid test, lane information cannot be guaranteed to be recovered properly even if the resolution of remote sensing photos has substantially improved.

### 3.3. Road Extraction from OpenStreetMap Files

The OpenStreetMap (OSM) [44] project is a knowledge collective that provides user-generated street maps, as shown in Figure 8b. The goal of OSM, which uses the peer production approach that led to the creation of Wikipedia, is to produce a set of map data that is open source, editable, and covered by new copyright laws. The main structure of OSM data is defined by three elements: nodes, ways, and relations. OSM supports exporting .osm map files, which can be converted to OpenDRIVE files (.xodr) by the conversion tool of VTD simulator, or to SUMO-supported map files (.net.xml) by the conversion tool of SUMO.

In current CAV tests, testing for individual functions often requires only a small section of road. For example, forward collision warning tests require different kinds of straight roads, and intersection collision warning tests require different kinds of intersections [45]. The road extracted by field collection is relatively single, and a small-scale collection cannot guarantee the coverage of the test. With the node information in the OSM file, intersections can be obtained by a hierarchical clustering algorithm, where adjacent nodes constitute an intersection [46,47]. This clustering model works directly at the semantic level. Compared to the pixel-level extraction work, this algorithm will save a lot of time. By processing and analyzing a large number of maps, the intersection library can be obtained and the coverage of the test can also be guaranteed.

Compared with field collecting and RS extraction, the method of extracting from OSM maps is less accurate, but efficient. In a short time, a large number of test roads can be obtained. Meanwhile, the related format conversion tools are more mature, and the conversion efficiency and effect are better.

## 4. Dynamic Scenario Generation

With the descriptions in Section 2 and Section 3, we can obtain the OpenDRIVE map file. Then, placing vehicles in the lanes of the map and setting the behavior, we can obtain the OpenSCENARIO file. The following forward collision warning test is used as the simplest example, as shown in Figure 9a. The remote vehicle (RV) is stationary in the tunnel, and the host vehicle (HV) approaches the RV at a speed of 60 km/h, as shown in Figure 9b. When the distance gets closer, until it is less than the minimum safe distance, the main vehicle will slow down or even change lanes, as shown in Figure 9c,d.

The map files can be obtained as mentioned in Section 3. Dynamic scenes can be generated by:1.Combinatorial testing;2.Knowledge-based generation;3.Driving behavior-based generation;4.Data-driven generation.

### 4.1. Combinatorial Testing

Combinatorial testing (CT) is a relatively common and simple method for generating scenarios from interactions between relatively few parameters. Using real-world acquisition data, hazard scenario cases [48,49,50,51] are reproduced or derived with the help of statistical model analysis. However, the probability of a vehicle encountering a hazard scenario while driving is often small and most are difficult to obtain directly from natural driving acquisitions, which makes it difficult to use statistical model derivation to adequately reach the safety boundaries of intelligent vehicle driving and requires analysis to explicitly describe and find the hazard boundaries and limits, which is precisely the challenge faced in current hazard scenarios.

Since manual testing cannot meet the demand for testing complex functions in virtual driving, Tatar et al. [52] developed TestWeaver software to automatically search the scenario parameter space to determine the system safety boundary. Testing is performed while generating scenarios and test reports to improve the testing efficiency.

Waymo has developed Carcraft [53], a virtual testing tool for autonomous driving technology, and Carcraft has become one of Waymo’s key technical capabilities. Special scenarios encountered in real car operations are repeated by modeling the scenarios in the virtual software Carcraft, and thousands of variants of the scenarios are derived in the virtual world using fuzzification methods. This approach enables the reproduction of problems encountered in real scenarios and provides preventive testing of similar scenarios that may occur, facilitating the analysis and resolution of chance events.

Xia et al. [54] proposed a method for creating Advanced Driver Assistance System (ADAS) scenarios. It is based on complexity index and combination testing, first defining various relevant factors affecting scenario generation, then setting specific weight values for each influencing factor, creating test cases by randomly combining the influencing factors and calculating the results. At the same time, they proposed a method to evaluate the generated scenarios based on a judgment matrix model. The complexity index (the sum of the weights of all influencing factors) of each test case is used to evaluate the performance of the test cases and guide the creation of better test cases. The effectiveness of the method is also verified in conjunction with a lane departure warning (LDW) test.

The following will briefly describe how to generalize to generate a Vulnerable Road User Collision Warning (VRUCW) [55] test scenario. Firstly, we need to determine the functional scenario: pedestrians cross the road at different locations with a speed of 1 m/s, the longitudinal distance between the host vehicle and the pedestrians is 20 m, and the host vehicle approaches the pedestrians at a different constant speed to test its warning capability, as shown in Figure 10a. Second, we determine the logical scenario: pedestrian position (5/12) m, host vehicle speed [10/20/30] km/h, as shown in Figure 10b. Third, we generate concrete scenarios and iterate through the combined parameters to obtain a total of six test scenarios, as shown in Figure 10c. Finally, the scenarios are formatted to form OpenSCENARIO files.

The basic concept of generating scenarios by CT:1.Generate functional scenarios. Determine the functions to be tested.2.Generate logical scenarios. Set the range of values and dispersion of the parameters of interest for the specific functional scenario.3.Generate concrete scenarios. According to the previously set parameter value range and dispersion, traverse the parameters to obtain the parameter combination of the scenario. At the same time, a filtering rule or direction of interest can be set during the traversal.4.Generate formatted scenes. Based on the tested simulator, generate supported scene file formats, such as OpenSCENARIO.

We formatted the scene and ran it at VTD, as shown in Figure 11. When the content to be tested is relatively simple or the parameter dimension is low, the CT method can quickly generate a large number of test scenarios and can cope with the initial testing.

From various angles, these works have supported the advancement of scenario-generating approaches, but fundamentally, they all use the concept of parameter traversal search to identify the system state space. After establishing a class of scenes, this approach is rather straightforward and may produce comparable scenarios rapidly and effectively. The traversal approach, however, finds it challenging to lock the test area of interest when the parameter range is wide, the dispersion is low, and there are more parameter types. This results in more repetitive scenes with a high degree of similarity, which has an impact on the effectiveness of the test.

### 4.2. Knowledge-Based Generation

The process of scenario CT has actually involved knowledge about scenarios, but this knowledge is fragmented and not systematically integrated. Meanwhile, in CAV testing practice, exhaustive testing is often impossible due to the exponential number of combinations [56]. To further address these issues, a holistic knowledge system needs to be constructed for scenario construction.

The W3C proposed the Web Ontology Language (OWL) [57] as a common way to handle the content of Web information and recommended it as the standard description language for the Semantic Web. With the continuous development and improvement of the Semantic Web technology system, researchers have started to apply Semantic Web technology to a wider range of application areas, such as geography, medicine, industry, military, etc. The role of ontology is to capture domain knowledge of the relevant domain and abstract a formal and machine-understandable model from the objective world [58,59,60,61].

Schuldt et al. [62] proposed a method for generating test suites from ontologies based on the automotive domain. Klueck et al. [63] proposed the use of an ontology to generate test suites for autonomous driving and autonomous driving functions, summarizing the relationship between ontology, test scenario generation, and test framework. Taking road generation as an example, they proposed the road section ontology, and a corresponding ontology transformation algorithm (CT ONT). The domains, variables, and constraints of lower-level concepts cumulatively form what is directly related to the higher-level concept. Thus, the conversion can be easily described as a recursive function. They both propose similar approaches, but Klueck’s approach allows automatic extraction of test suites directly from the ontology, providing convenience for testing.

Tao et al. [64,65] proposed an ontology-based test scenario generation method for CAV, describing the specific flow from ontology to CT to test simulator (VTD). Taking Autonomous Emergency Braking (AEB) as an example, an ontology will be constructed for the Autonomous Emergency Braking (AEB) functional prototype of AVL List GmbH, and then the ontology will be converted into its corresponding CT input model with explicit constraints, parameters, and their values. The test cases generated from the CT will then be used for virtual simulations, verifying the feasibility of the solution.

Steimle et al. [66] further summarize the definitions, concepts, and frameworks in CAV testing. They propose and describe terms that are particularly relevant to the overview of scenario-based approaches to the development and testing of autonomous vehicles. These terms represent a basic glossary that will be expanded in the future. The terms and their relationships at each stage are visualized as UML diagrams (including scenarios, scenario definitions, test flows, etc.). This visualization enables quick identification of the relationships and dependencies between the different terms. Furthermore, Adaptive Cruise Control (ACC) testing is used as an example to subdivide the ACC system into different elements, illustrating the terminology parts and software units such as system, components, hardware, and software units.

Here, we also use Vulnerable Road User Collision Warning (VRUCW) as an example to briefly describe how to generate relevant test scenarios through the ontology.

First, we organize the parameters related to statistical testing and describe the scenario as a logical scenario, as shown in Figure 12.

Then, we organize the mapping relationships between parameters and the quantitative relationships to represent this ontology in UML, as shown in Figure 13. It should be noted that buildings are not required in the VRUCW test, so the interval is [0,N]. With the ontology transformation algorithm, the ontology model can be used as input and the test scene library can be output [62].

The above work describes in detail the knowledge areas and related concepts involved in the CAV testing process and constructs an ontology to automate the generation of scenarios. The ontology construction of the simulation test scenarios further discusses the components of the scenarios and the organization network, providing a theoretical basis for the machine learning-based scenario generation method. Knowledge-based scenario generation can effectively increase the coverage of generated scenarios. However, the essence of its method is still generalized generation, which cannot lock the key scenes without combining filtering rules. Moreover, the vehicles in the scenes generated using these methods do not incorporate a motion model, so some scenes may be generated that do not conform to real-world kinematics.

### 4.3. Driving Behavior-Based Generation

Relying on the researcher’s understanding and experience of the changing patterns within the transportation system, it is possible to obtain a motion model that describes the patterns and is applicable on a large scale [67]. These models can be used in the field of scenario generation.

The basic driving behavior consists of Car-following and Lane-changing, as shown in Figure 14. Where the yellow car is the remote vehicle (RV) and the white car is the host vehicle (HV). The following describes the methodology for generating tests based on driving behavior:1.Set the motion state of the RV, such as uniform speed, or decelerate driving.2.Simulate the driving of the HV using a driving model, such as the car-following model.3.Adjust the driving model or model parameters, such as aggressive and conservative, start testing, and record vehicle data.4.Convert the vehicle data into an OpenSCENARIO file and repeat the previous step.

We will introduce the commonly used Car-following model. Car-following behavior, which characterizes a vehicle’s conduct when it is following another vehicle in front of it, has a substantial influence on traffic performance, safety, and air pollution. Furthermore, vehicle following behavior is a critical component of microscopic simulation models [68].

The intelligent driver model (IDM) [69,70] describes a typical following behavior. The model combines the desired speed, acceleration performance, following distance, and driver’s habit during vehicle driving, and is a commonly used following model in various simulators, such as SUMO. The IDM is also commonly used in CAV-related fields as a common driving model [71,72,73].

The Extended Intelligent Driver Model (EIDM) [74] is built on various well-known model extensions of IDM. It also incorporates computations to lessen jerking in various driving scenarios (lane changes, accelerating from a standstill, etc.). Each extension may be separately turned “off” by adjusting the parameters (mostly to 0). The model’s goal is to accurately mimic single vehicle and driver submicroscopic acceleration characteristics.

The safety distance model (Krauss model) [75,76] is another commonly used Car-following model, which is also widely used in SUMO [77]. The main feature of this model is the parameters describing the typical acceleration and deceleration capabilities of the vehicle. Real vehicles occasionally break the safe distance rule and, on average, the gap for real vehicles is slightly smaller than in the Krauss model. From a driving style perspective, Tan et al. [78] proposed an aggressive following model. Simulation results show that the new model can simulate an aggressive driving style, which is important for simulating traffic using different driving style models.

Lane-changing models are also a common vehicle motion model [79] and classified as either mandatory (MLC) or discretionary (DLC) [80]. A mandatory lane change is made when the driver must leave the current lane. DLC is used to improve driving conditions. Daniel Krajzewicz et al. [81] developed the lane change model DK2008. Based on DK2008 optimization formed LC2013 [82] summarizes four different lane change motives: strategic lane change, coordinated lane change, tactical lane change, and obligatory lane change, which are commonly used lane change models today. By setting RV’s behaviors (e.g., stationary and braking) and the lane change model, test scenarios such as overtaking can be generated.

The following is a brief description of IDM model and the parameters in the model are shown in Table 3. The speed difference between the HV and RV:(1)$$\Delta v=v_{rv}-v_{hv}$$

Their desired following gap:(2)$$g\left(v_{hv},\Delta v\right)=G_{0}+max\left(0,v_{hv}\cdot T+\frac{v_{hv}\cdot \Delta v}{2\sqrt{A_{max}\cdot D_{com}}}\right)$$

When Δv→0, the desired following gap:(3)$$g\left(v_{hv},0\right)=G_{0}+v_{hv}\cdot T$$

According to Equations (Equation 1)–(Equation 3), the acceleration of the HV:(4)$$a_{hv}=A_{max}\left[1-{\left(\frac{v_{hv}}{V_{e}}\right)}^{\beta }-{\left(\frac{g\left(v_{hv},\Delta v\right)}{G_{0}}\right)}^{2}\right]$$

The first part $A_{max}\left(1-{\left(\frac{v_{hv}}{V_{e}}\right)}^{\beta }\right)$ indicates an appropriate acceleration rate toward the expected speed $v_{e}$. The latter part $-A_{max}{\left(\frac{g\left(v_{hv},\Delta v\right)}{G_{0}}\right)}^{2}$ represents a braking strategy according to the current gap and the desired minimum gap $G_{0}$ to the RV.

In summary, the control behavior of the HV can be given by Equation (Equation 4) when the RV is traveling at a certain speed and the HV is following the RV.

Based on the Car-following model, many scenarios can be generated, such as forward collision warning (FCW), emergency braking vehicle warning (EBW), etc. The Lane-change model and Cut-in model can also be used to generate test scenarios. The performance of the tested algorithm (FCW/EBW) can be verified while generating scenarios, which can improve the testing efficiency. The advantage of the driving behavior-based scenario generation method is that it combines the driving model to ensure that the generated scenarios are following the laws of motion. At the same time, the scenes generated in this way can ensure that the scenes generated have criticality and can avoid the generation of meaningless scenes (e.g., the distance between two cars is getting farther and farther apart). At the same time, different types of test scenes can be generated flexibly by adjusting the parameters of the driving model.

However, there are some limitations to this approach. The driving behavior of RV needs to be set in advance by humans. Once the behavior of a certain car is fixed, the generated scenes have one less dimension and the coverage of the test scenes cannot be guaranteed. Furthermore, the method can only cover the dynamic scene part, and the static scene elements still need to be combined with the CT method.

### 4.4. Data-Driven Generation

Driving behavior modeling can describe the motion patterns of vehicles. On the one hand, multiple CAVs with similar motion models ignore the randomness of vehicle motion. On the other hand, when generating a large number of scenes, similar motion models lead to increased repetition of scenarios. Since real data contains a large amount of behavioral characteristic information (e.g., driving habits, driving memories), some researchers have used a data-driven approach to study the generation of simulation test scenarios.

Wei et al. [83] proposed a self-learning support vector regression (SVR) modeling method to study the asymmetric characteristics of the following and its effect on the evolution of traffic flow. The current moment’s motion state (velocity, relative velocity) is input and the next moment’s motion behavior (velocity) is output.

Xie et al. [84] proposed a deep learning-based autonomous lane changing model for traffic vehicles, combining deep belief network (DBN) and long short-term memory (LSTM) neural network, using next generation simulation (NGSIM) [85] data to build two modules for lane changing decision and lane changing execution and integrating the model to accurately predict the lane changing process of vehicles and explore the basic features of lane changing behavior.

With the development of machine learning, reinforcement learning [86,87,88] has also started to be applied to the field of scenario generation. Feng et al. [89,90,91] proposed a test scenario library generation (TSLG) framework for solving automatic scenario generation problems using reinforcement learning, and described the detailed generation process with the actual problems of vehicle cut-in, lane changing, and car following, providing a good summary and outlook for the field of test scenario generation. The scenario generation problem is described as a Markov Decision Process (MDP) [92] problem, and the scenario of interest is generated by solving it in Q-learning.

In this paper, we present the methodology for generating scenes based on MDP. During the test scenario generation process, we can define a test scenario as:(5)$$x={\left[s_{0},a_{1},a_{2},\ldots ,a_{n}\right]}^{T},x\in X$$
where $s_{0}$ denotes the initial state of the environment, $s_{0}$ may contain information about the host vehicle and the remote vehicle, etc., *n* denotes the total number of frames in the scene, $,a_{1},a_{2},\cdots ,a_{n}$ represents the action series of the vehicles in the environment. The action contains some control behaviors of the car, such as acceleration, deceleration, lane change, etc.

Its basic reward function is defined as:(6)$$R=\left\{\begin{array}{cc} 1, & s\in X_{i}\\ -1, & s\notin X_{i} \end{array}\right.$$
where $X_{i}$ denotes the scenario of interest, i.e., the test intent. Choose the action (a) that maximizes $Q(S^{\prime },a)$ as the next moment’s action ($A^{\prime }$) to update the value function:(7)$$Q\left(S,A\right)=Q\left(S,A\right)+\alpha \left(R+\gamma \max_{a}Q\left(S^{\prime },a\right)-Q\left(S,A\right)\right)$$
where $S^{\prime }$ is the new state, α is the learning step, and γ is the decay factor. If $S^{\prime }$ is the termination state, the iteration is completed.

Take the FCW test scenario as an example, its MDP model is shown in Figure 15. Where state is:(8)$$S=(v_{rv},\Delta v,g)$$

The action is the deceleration of the RV, and the rear vehicle follows using the IDM model. When the test intent is a hazard scenario (imminent collision), the reward can be set as:(9)$$R=\left\{\begin{array}{cc} 1, & g<1m\\ -1, & g⩾1m \end{array}\right.$$

So far, the MDP modeling is completed, and many methods can solve the problem, such as Q-table, Deep Q Network (DQN), etc. Through the above methods, multiple sets of action sequences can be obtained, and each set of action sequences with initial states constitutes a test scenario. The scenarios generated using reinforcement learning can closely match the test intent.

However, the current scenario generation methods still have a problem in that they do not incorporate sensor errors. Therefore, the constructed scenarios are relatively accurate in terms of the vehicle’s position. Although the sensor errors can be simulated by the simulator, this introduces a lot of overhead and increases the burden of testing. Therefore, it is a more convenient way to introduce errors in the scenarios in the simulation tests.

Of all the methods mentioned in this paper, the combined test approach can introduce errors quite simply. However, the traditional work does not take into account the sensor error, and usually, the default sensor has a certain stability. However, in the actual test, we find that often there may be data loss due to communication problems. Based on the MDP generation scenario, it is also considered to be the case that the sensor is working properly. We can describe the test scenario generation problem as a Partially Observable Markov Decision Process (POMDP) problem. POMDP is an ideal model for sequential decision-making in dynamic uncertain environments where the state of the environment is partially knowable, with the central point that the agent cannot know the state of the environment it is in and needs to resort to additional sensors. It is an important branch of stochastic decision process research, which can objectively and accurately describe the real world.

## 5. Conclusions

As a new technology, CAV not only provides a more comfortable and safer traffic environment but is also important for improving traffic efficiency, reducing pollution, and lowering accident rates. CAV testing is also an essential part of the process. As real field testing is time-consuming and inefficient. Simulation testing has advantages such as high efficiency and automation. Therefore, the current mainstream practice is to carry out simulation testing first, to ensure passing of the simulation test before carrying out the actual field test. The core part of simulation testing is the test scenario library. The method of manually building the scenario library is also inefficient, so the issue of automated scenario generation has attracted the attention of the industry.

First, we organized the relevant definitions used in simulation testing. The scenarios are divided into functional scenarios, logical scenarios, concrete scenarios, and formatted scenarios. The support for the relevant formats is also given in conjunction with the common simulators. The scenario generation we discuss is the generation of concrete scenarios as well as formatted scenarios. Scenarios mainly include static maps and dynamic scenarios.

Second, we compared the current methods of static map generation, as shown in Table 4. Collecting by collecting vehicles is the most real-time as well as accurate method, but it is not efficient and difficult to cover a large area. As the resolution of remote sensing maps has increased, the method of extracting roads using remote sensing has become more common. This method can quickly extract a large area map, but the lane information may be missing and the effectiveness cannot be guaranteed. Finally, we describe the method of extracting roads using OSM, which ensures efficiency and includes many road details (e.g., lane information, etc.). Therefore, in CAV testing, to ensure testing efficiency, roads can be extracted and classified by OSM to build a test road library.

Finally, we introduce the method of dynamic scenario generation, as shown in Table 5. Based on a static map, dynamic vehicles or pedestrians need to be placed to constitute dynamic scenes. We divide the current methods into four categories. The first category is the method of combinatorial testing, which can achieve rapid scene generation by arranging and combining parameters. This method is efficient, but the generated scene quality availability is low and there may be irrational scenes. The second type of method is the knowledge-driven scenario generation method, which is mainly used to generate scenarios by constructing the ontology of scenarios. This method is a further summary of the CT method, which can further organize the parameters and logical relationships in dynamic scenes. At the same time, the ontology construction provides a theoretical basis for generating scenes by machine learning. The third type of method is to model the motion behavior of vehicles for scene generation. The scenes generated by this method are relatively more in line with the laws of motion, but lack randomness and are more difficult to cover all test contents. The fourth method is data-driven, taking the historical motion state of the vehicle as input and outputting the motion state of the next moment. This method can closely match the test intent and conforms to the motion law.

In future research on scenario generation methods, we believe that data-driven generation methods can be used to generate scenarios. On this basis, sensor errors are introduced to build more realistic test scenarios. 

## Figures and Tables

**Figure 1 sensors-22-07735-f001:**
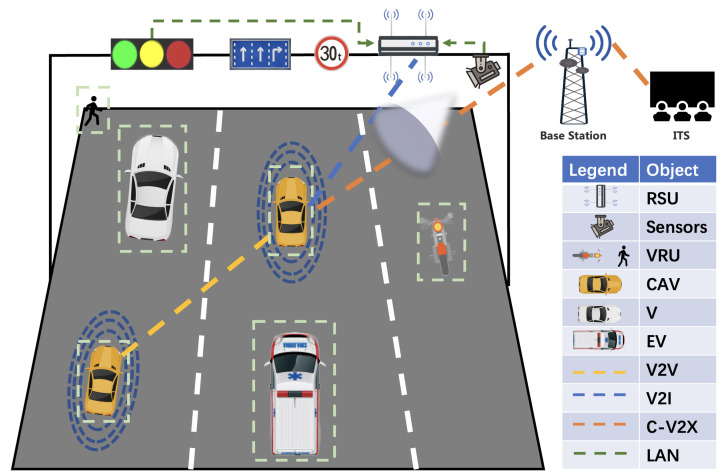
How connected and automated vehicle works and other traffic participants.

**Figure 2 sensors-22-07735-f002:**
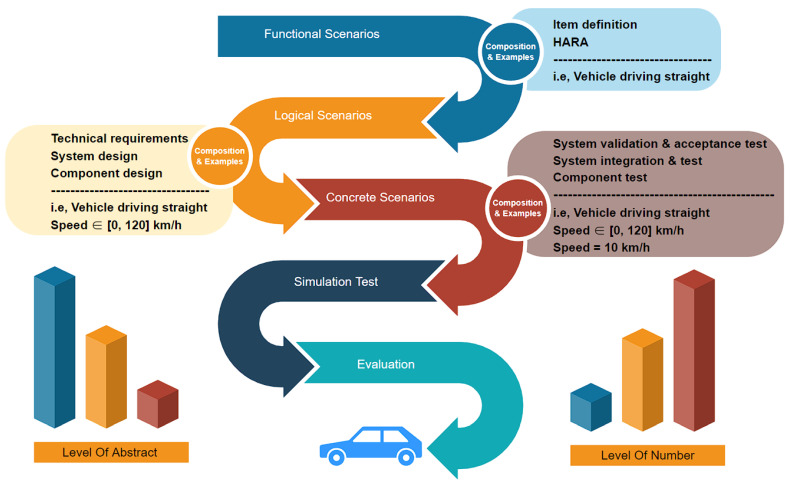
The “function-logic-concrete” three-tier hierarchy of scenarios and the testing process; HARA: hazard analysis and risk assessment.

**Figure 3 sensors-22-07735-f003:**
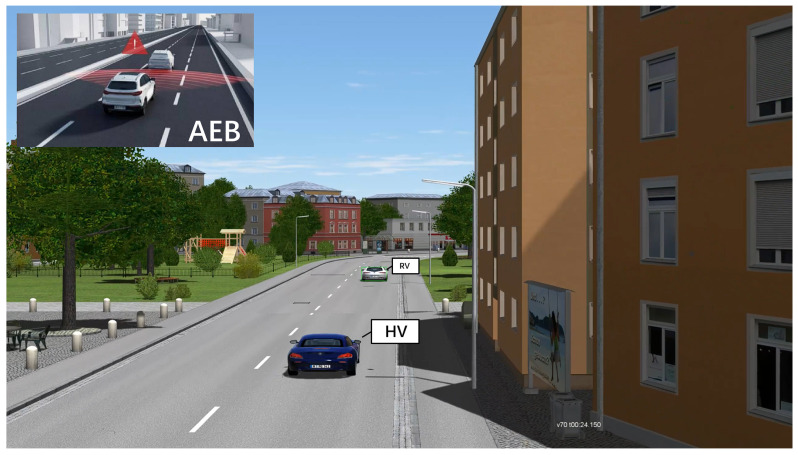
Autonomous Emergency Braking (AEB) function test.

**Figure 4 sensors-22-07735-f004:**
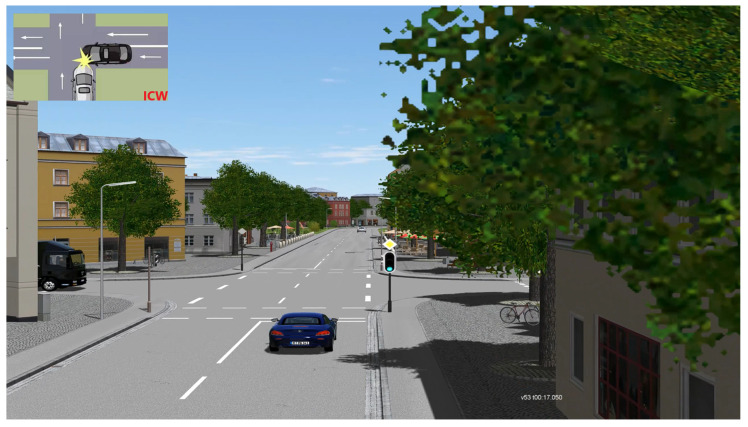
Intersection Collision Warning (ICW) function test.

**Figure 5 sensors-22-07735-f005:**
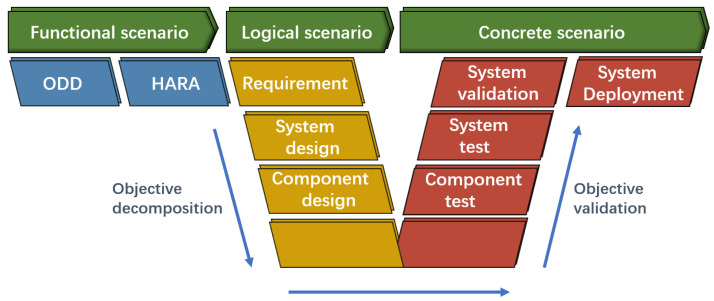
Functional testing scenarios during V-model-based development process.

**Figure 6 sensors-22-07735-f006:**
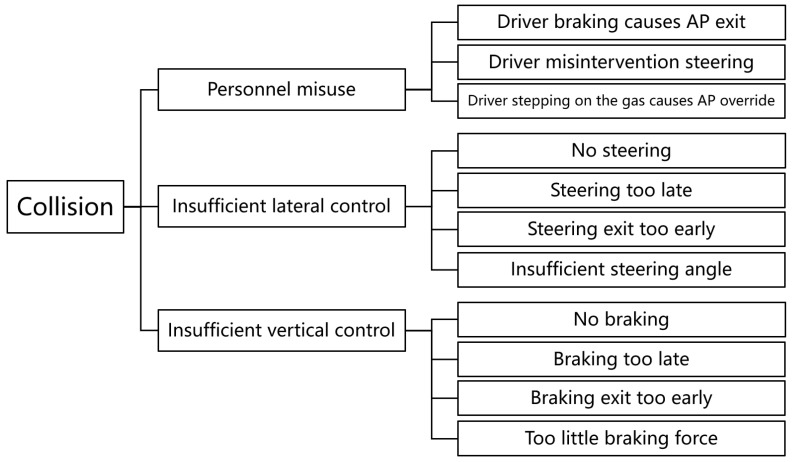
Vehicle-level hazards in the cut-in functional testing.

**Figure 7 sensors-22-07735-f007:**
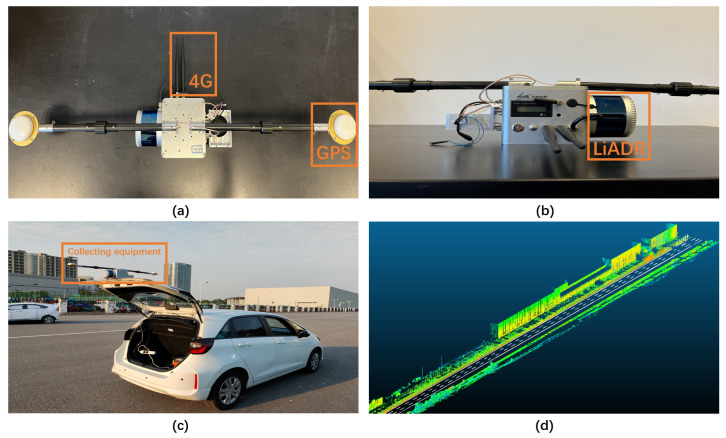
(**a**) Top view of portable map collecting equipment; (**b**) main view of portable map collecting equipment; (**c**) vehicles equipped with collection equipment; (**d**) collected point cloud image.

**Figure 8 sensors-22-07735-f008:**
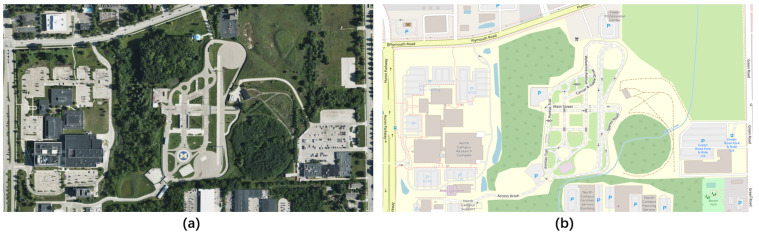
(**a**) Remote sensing images; (**b**) OpenStreetMap maps.

**Figure 9 sensors-22-07735-f009:**
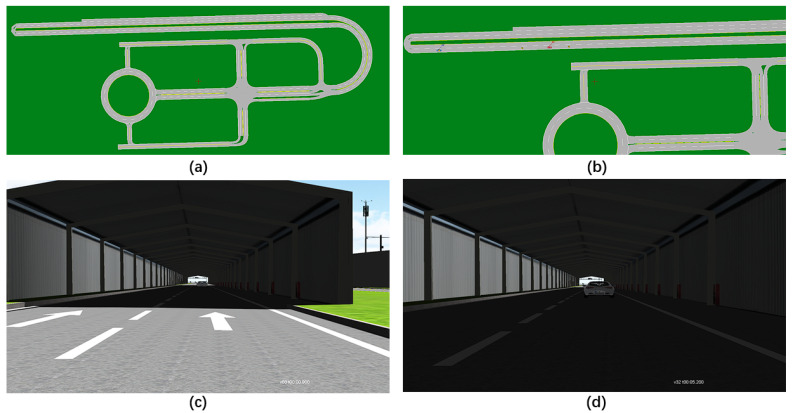
(**a**) Closed test field map in OpenDRIVE Odr Viewer. (**b**) OpenSCENARIO editing interface for forward collision warning test scenarios in VTD simulator. (**c**) Effect of forward collision warning test (HV’s speed = 60 km/h time = 0.9 s) in VTD simulator. (**d**) Effect of forward collision warning test (HV’s speed = 32 km/h time = 5.2 s) in VTD simulator.

**Figure 10 sensors-22-07735-f010:**
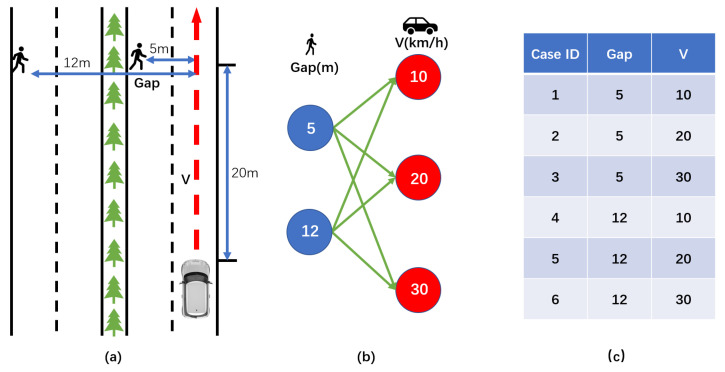
(**a**) Functional scenario description. (**b**) Parameter ranges for logical scenarios. (**c**) List of key parameters for the 6 concrete scenarios generated.

**Figure 11 sensors-22-07735-f011:**
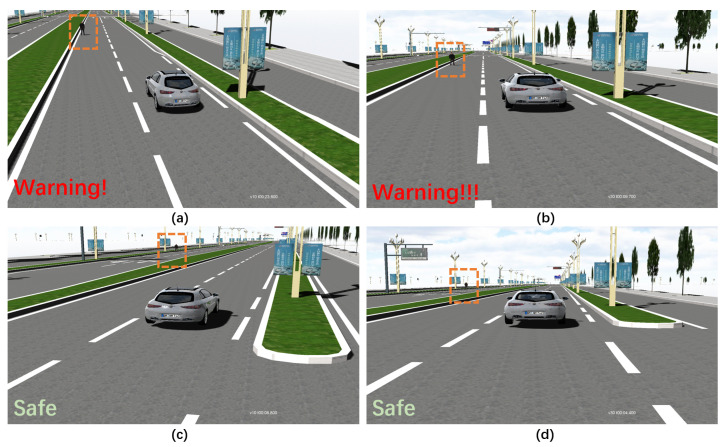
VRUCW test in VTD simulator: (**a**) case 1: Gap = 5 m, HV’s speed = 10 km/h; (**b**) case 2: Gap = 5 m, HV’s speed =30 km/h; (**c**) case 3: Gap = 12 m, HV’s speed = 10 km/h; (**d**) case 4: Gap = 12 m, HV’s speed = 30 km/h.

**Figure 12 sensors-22-07735-f012:**
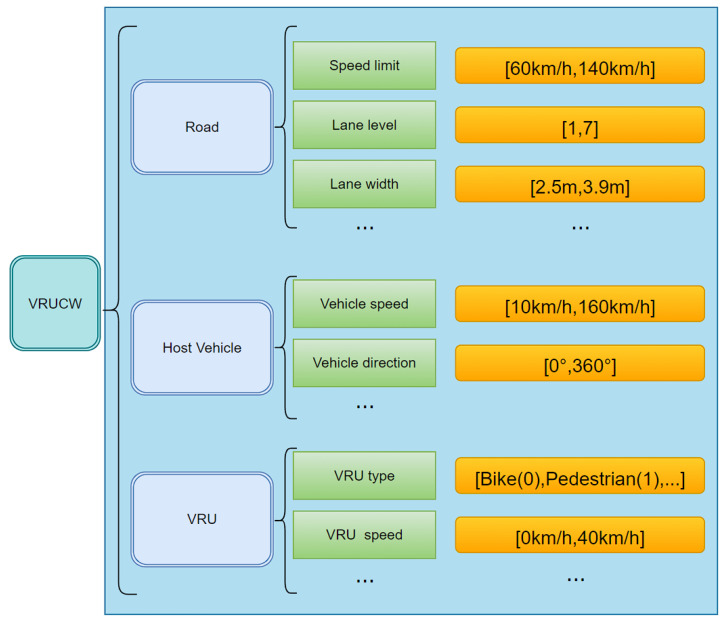
Logical scenarios for VRUCW.

**Figure 13 sensors-22-07735-f013:**
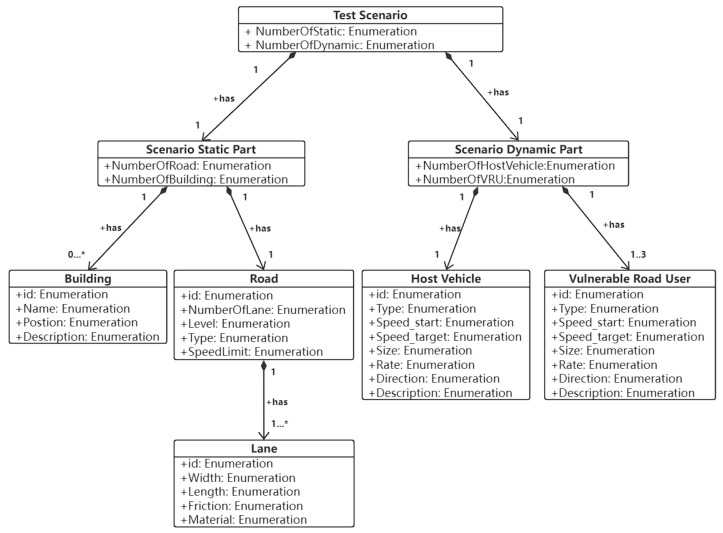
Constructed VRUCW ontology using Unified Modeling Language (UML).

**Figure 14 sensors-22-07735-f014:**
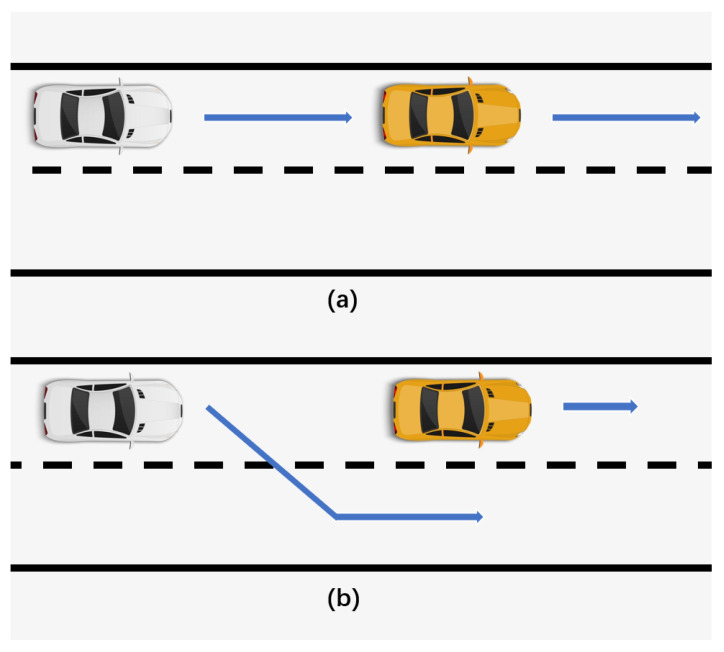
Basic driving behavior, (**a**) car-following, (**b**) lane-changing.

**Figure 15 sensors-22-07735-f015:**
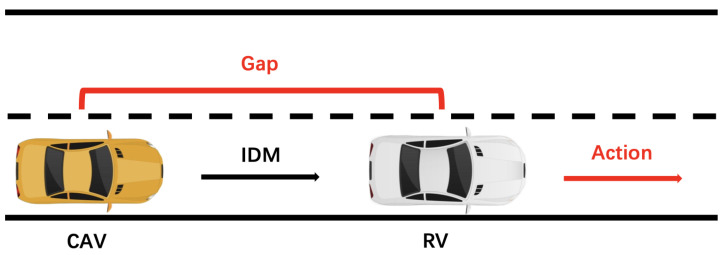
Car-following’s MDP model.

**Table 1 sensors-22-07735-t001:** V2X functionality test content.

Category	Full Name
P2X	Vulnerable Road User Safe Passing
V2I	Hazardous Location Warning
V2I	Speed Limit Warning
V2I	Red Light Violation Warning
V2I	Green Light Optimal Speed Advisory
V2I	In-Vehicle Signage
V2I	Traffic Jam Warning
V2I	Vehicle Near-Field Payment
V2I	Cooperative Vehicle Merge
V2I	Cooperative Intersection Passing
V2I	Differential Data Service
V2I	Dynamic Lane Management
V2I	Cooperative High Priority Vehicle Passing
V2I	Guidance Service in Parking Area
V2I	Probe Data Collection
V2I	Road Tolling Service
V2P/V2I	Vulnerable Road User Collision Warning
V2V	Forward Collision Warning
V2V	Blind Spot Warning-Lane Change Warning
V2V	Do Not Pass Warning
V2V	Emergency Vehicle Warning
V2V	Cooperative Platooning Management
V2V/V2I	Intersection Collision Warning
V2V/V2I	Left Turn Assist
V2V/V2I	Sensor Data Sharing
V2V/V2I	Cooperative Lane Change
V2V-Event	Emergency Brake Warning
V2V-Event	Abnormal Vehicle Warning
V2V-Event	Control Loss Warning

**Table 2 sensors-22-07735-t002:** Scenario file support for common simulators.

Simulators	OpenDRIVE	OpenSCENARIO
CARLA	yes	yes
CarMaker	yes	Yes
CarSim	yes	No
MATLAB	yes	No
PanoSim	yes	Yes
PreScan	yes	Yes
PTV Vissim	yes	No
SUMO	yes	No
VIRES VTD	yes	Yes

**Table 3 sensors-22-07735-t003:** IDM model parameter notation.

Parameters	Notation	Value
$A_{max}$	Maximum acceleration	2 m/s${}^{2}$
$D_{com}$	Comfortable deceleration	3 m/s${}^{2}$
$V_{e}$	Expected speed of HV	60 km/h
*T*	Desired safety time headway	1 s
β	Acceleration factor	2
$G_{0}$	Minimum safe gap	1 m
*g*	Gap between HV and RV	-
$v_{hv}$	Speed of HV	-
$v_{rv}$	Speed of RV	-

**Table 4 sensors-22-07735-t004:** A comparison of different road extraction methods.

Methods	Efficiency	Coverage	Accuracy	Real-Time
Actual field collection	Moderate	Moderate	good	good
Remote sensing imagery	good	good	Moderate	Moderate
OpenStreetMap files	good	good	good	Moderate

**Table 5 sensors-22-07735-t005:** A comparison of different dynamic scenario generation methods.

Methods	Ease of Use	Conform to Driving Rules	Percentage of Key Scenarios	Quality
Combinatorial testing	good	Moderate	Moderate	Moderate
Knowledge-based generation	Moderate	Moderate	Good	Moderate
Driving behavior based generation	Moderate	good	Moderate	good
Data-driven generation	Moderate	good	good	good

## Data Availability

Not applicable.

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
