# Peer review of "Review on Functional Testing Scenario Library Generation for Connected and Automated Vehicles"

_sensors, 2022, doi:10.3390/s22207735_

Round 1

Reviewer 1 Report

This paper reviews techniques to construct test scenarios for self-driving cars and based on that, propose their own test case generation approaches. The main problem with this paper is that the purposes of testing are not clearly identified. In reality, the ultimate goals of testing are to ensure that the CAV can meet some essential security/safety standards and best practices before commercialisation. The goals will lead to some requirements that the test generations should rely on. The testing scopes can be varied, from individual sensor to internal communication protocols or cooperation between vehicles on the road. However, the testing scope in this paper is not well-defined. There are also some underlying test models such as the V model and some widely used methods such as FMEA, FTA, but I can’t find their position in the authors’ work. Overall, the review in this paper has not been conducted systematically, therefore it is neither comprehensive nor coherent. The authors also proposed their own method based on previous work, but did not highlight the differences and did not show evidence of improvement, therefore these contributions are unconvinced. 

Author Response

Dear Editors and Reviewers:

Thank you for your letter and for the reviewers’ comments concerning our manuscript entitled “Review on Testing Scenario Library Generation for Connected and Automated Vehicles” (ID: sensors-1902590). Those comments are all valuable and very helpful for revising and improving our paper, as well as the important guiding significance to our research. We have studied comments carefully and made corrections, which we hope meet with approval. The main corrections in the paper and the responses to the reviewer’s comments are as flowing:

Point 1: This paper reviews techniques to construct test scenarios for self-driving cars and based on that, propose their own test case generation approaches. The main problem with this paper is that the purposes of testing are not clearly identified. In reality, the ultimate goals of testing are to ensure that the CAV can meet some essential security/safety standards and best practices before commercialisation. The goals will lead to some requirements that the test generations should rely on. The testing scopes can be varied, from individual sensor to internal communication protocols or cooperation between vehicles on the road. However, the testing scope in this paper is not well-defined. There are also some underlying test models such as the V model and some widely used methods such as FMEA, FTA, but I can’t find their position in the authors’ work. Overall, the review in this paper has not been conducted systematically, therefore it is neither comprehensive nor coherent. The authors also proposed their own method based on previous work, but did not highlight the differences and did not show evidence of improvement, therefore these contributions are unconvinced.

Response 1: We are very sorry that we missed some important descriptions. The tests mentioned in this study are mainly functional tests, which we will highlight in the title as well as in the abstract. Also, we will highlight our test content in the body of the text. These test contents are dependent on the test scenarios, and this study will focus on describing the contents related to the construction of the test scenarios. However, these studies will not emphasize the test methods and test results. This study mainly describes the current test scenario generation methods. I also give a brief example of each generation method to provide a reference for researchers to choose a generation method. We do not propose a new test method, but only a summary of the current methods. But we also present the idea that road generation methods and dynamic traffic participants need to be closely integrated. In previous studies, road generation and traffic participant generation have often been separated. But current tests need to closely integrate test roads and traffic participants.

Thank you very much for your constructive comments and suggestions, which have helped us a lot to improve the quality of the paper, both in terms of English and depth. We have tried our best to improve the issues that still need improvement and have made some changes in the manuscript. We sincerely thank the editors/reviewers for their enthusiastic work and hope that these changes will be recognized.

Reviewer 2 Report

It was exciting to review this manuscript. Here are a few comments to improve it.

The major issue with this study is the fact that it assumes that the open road test did not start with simulation analysis. To my understanding, before traditional open road test or closed field test, simulation analysis have already been done. The key question is does this study provide new insights that have not been established in the previous simulation analysis? I think the study should lay out these new  findings in the abstracts and consistently build their cases based on the manuscript

A review of  the previous studies that focused on open test AVs safety (line 49-55) would be very interesting if the study could explore the actual shortfalls of AVs. For instance, when does the AV is likely to be at fault? or how does the AV operation affect traffic flow when the AV interact with pedestrians? That way, it could be a better reason to show that there are still some issues associated with the AV operations that could be solved by additional simulatio analysis. 

Author Response

Dear Editors and Reviewers:

Thank you for your letter and for the reviewers’ comments concerning our manuscript entitled “Review on Testing Scenario Library Generation for Connected and Automated Vehicles” (ID: sensors-1902590). Those comments are all valuable and very helpful for revising and improving our paper, as well as the important guiding significance to our research. We have studied comments carefully and made corrections, which we hope meet with approval. The main corrections in the paper and the responses to the reviewer’s comments are as flowing:

Point 1: The major issue with this study is the fact that it assumes that the open road test did not start with simulation analysis. To my understanding, before traditional open road test or closed field test, simulation analysis have already been done. The key question is does this study provide new insights that have not been established in the previous simulation analysis? I think the study should lay out these new findings in the abstracts and consistently build their cases based on the manuscript.

Response 1: This understanding is very accurate, before traditional open road test or closed field test, simulation analysis have already been done. The main work of this study integrates and discusses related methods such as road generation and dynamic scene generation. Previous work tended to always describe how to generate roads or how to generate dynamic scenarios separately. But current tests require a close integration of roads and traffic participants. Therefore, this study integrates the two works and provides a classification of the existing methods as well as a simple example of the existing classical methods. In particular, this study describes how reinforcement learning can be used to generate scenarios. This is one of the more popular areas of research at the moment, and the examples provide a basis for researchers to study.

Point 2: A review of the previous studies that focused on open test AVs safety (line 49-55) would be very interesting if the study could explore the actual shortfalls of AVs. For instance, when does the AV is likely to be at fault? or how does the AV operation affect traffic flow when the AV interact with pedestrians? That way, it could be a better reason to show that there are still some issues associated with the AV operations that could be solved by additional simulatio analysis.

Response 2: Again, I think this is a very interesting question and I will add relevant content to this study. We are not only looking at functional test case generation. We are also working on V2X-HIL testing and have published papers on it as well. In the actual testing, we have encountered the problem of inaccurate/untimely GPS positioning. When testing the warning function, we found that some of the tested parts (OBU) did not consider the elevation of the vehicle, and when the vehicle was located near an overpass, the position resolution of the vehicle would be wrong (spatial misalignment), leading to false triggering of the warning.

Thank you very much for your constructive comments and suggestions, which have helped us a lot to improve the quality of the paper, both in terms of English and depth. We have tried our best to improve the issues that still need improvement and have made some changes in the manuscript. We sincerely thank the editors/reviewers for their enthusiastic work and hope that these changes will be recognized.

Reviewer 3 Report

1. Please include the word 'functional' in the title as this paper is about functional testing and not on all types of testing. 

2. Please use a running example throughout the paper to explain the concepts in depth. Start the running example early on, say after the second paragraph in Section 2.

3. Must add more examples and elaborate explanation of more use cases to enrich the paper. 

4. Must include a discussion of the CARLA simulator as well and discuss whether its integration with CarSim covers openDRIVE and openSCENARIO use cases. Similarly, must include the discussion of Matlab's Autonomous Vehicle toolbox. Also, if possible, include a discussion on Eclipse MOSAIC. 

5. In the first sentence of the first paragraph in Section 4:

Chapter 2 and Chapter 3  -------->  Section 2 and Section 3 

Author Response

Dear Editors and Reviewers:

Thank you for your letter and for the reviewers’ comments concerning our manuscript entitled “Review on Testing Scenario Library Generation for Connected and Automated Vehicles” (ID: sensors-1902590). Those comments are all valuable and very helpful for revising and improving our paper, as well as the important guiding significance to our research. We have studied comments carefully and made corrections, which we hope meet with approval. The main corrections in the paper and the responses to the reviewer’s comments are as flowing:

Point 1: Please include the word 'functional' in the title as this paper is about functional testing and not on all types of testing.

Response 1: I have changed the title of the article to "Review on Functional Testing Scenario Library Generation for Connected and Automated Vehicles".

Point 2: Please use a running example throughout the paper to explain the concepts in depth. Start the running example early on, say after the second paragraph in Section 2.

Response 2: Your suggestion is very good and we have developed a description of the concept in the context of AEB.

One of the more critical features examined in the automatic driving functional test is the Autonomous Emergency Braking (AEB) . The AEB functioning scenario is as follows: when the host vehicle (HV) senses a potential accident, the car immediately brakes, assuring the driver's safety. When the HV is moving in a straight line and the safety distance from the remote vehicle (RV) is short, the AEB feature is activated, as shown in Figure 1.

Taking the Autonomous Emergency Braking (AEB) as an example, the recommended operating range of AEB in urban areas is [0,70] km/h. The definition of parameter ranges for logical scenarios can clarify the scope of the test, avoid meaningless testing, and improve the efficiency of the test.

Again, using AEB as an example, its concrete scenario will specifically describe the speed of the vehicles, the spacing between the vehicles, etc. For example, the host vehicle speed in Figure 1 is 70km/h and the spacing between the two vehicles is 30m at 24.150s .

Point 3: Must add more examples and elaborate explanation of more use cases to enrich the paper.

Response 3: We have added a specific description of the AEB, ICW test cases.

Point 4: Must include a discussion of the CARLA simulator as well and discuss whether its integration with CarSim covers openDRIVE and openSCENARIO use cases. Similarly, must include the discussion of Matlab's Autonomous Vehicle toolbox. Also, if possible, include a discussion on Eclipse MOSAIC.

Response 4: CARLA is an open-source simulator for autonomous driving research that also supports OpenDRIVE and OpenSCENARIO formats. CarSim supports the OpenDRIVE format but does not directly support the OpenSCENARIO format. CARLA integrated CarSim support OpenDRIVE and OpenSCENARIO formats. Matlab's Autonomous Vehicle toolbox (Driving Scenario Designer) also supports OpenDRIVE, but not OpenSCENARIO format. In the case of Eclipse MOSAIC, we have not used it directly, so we won't discuss it here.

Point 5: In the first sentence of the first paragraph in Section 4:

Chapter 2 and Chapter 3  -------->  Section 2 and Section 3

Response 5: It was an oversight on our part and has been corrected.

Thank you very much for your constructive comments and suggestions, which have helped us a lot to improve the quality of the paper, both in terms of English and depth. We have tried our best to improve the issues that still need improvement and have made some changes in the manuscript. We sincerely thank the editors/reviewers for their enthusiastic work and hope that these changes will be recognized.

Round 2

Reviewer 1 Report

After revision, this paper still doesn't look like a good coverage of functional testing approaches to me. In my opinion, with the scenarios generated based on the approaches reviewed in this paper, the test cannot make any reliable statement about the functional (safety) of the CAV system. The paper misses some underlying requirements for functional testing, for example there should be (many) scenarios considering the consequences if one or some of the sensors (lidar, radar or GPS) stops working. Therefore, it is important to review Failure Modes and Effects Analysis (FMEA) approaches. For road generation there should also be some scenarios that consider the conditions affecting the sensors operation (e.g. weather conditions like light, rain, snow; the occurrences of material affecting the functionalities of the sensors – e.g. Lidar is known to be inaccurate when detecting mesh objects; or the occurrences of environmental noise or security attacks such as jamming). To avoid overlooking many essential test scenarios, authors should systematically review the problem based on a widely accepted test model (e.g. the V-model) and focus on the requirements (for each development phases) and examine the approaches that generate the scenarios to meet these requirements.

Author Response

Dear Editors and Reviewers:

Thank you very much for your constructive comments and suggestions, please check the attachment.

Reviewer 3 Report

The authors have satisfactorily answered all comments. 

Author Response

Dear Editors and Reviewers:

Thank you for your letter and for the reviewers’ comments concerning our manuscript entitled “Review on Functional Testing Scenario Library Generation for Connected and Automated Vehicles” (ID: sensors-1902590). Those comments are all valuable and very helpful for revising and improving our paper, as well as the important guiding significance to our research. We have studied comments carefully and made corrections, which we hope meet with approval. The main corrections in the paper and the responses to the reviewer’s comments are as flowing:

Point 1: The authors have satisfactorily answered all comments.

Response 1: Thank you very much for your constructive comments and suggestions, which have helped us a lot to improve the quality of the paper, both in terms of English and depth. We have tried our best to improve the issues that still need improvement and have made some changes in the manuscript. We sincerely thank the editors/reviewers for their enthusiastic work and hope that these changes will be recognized.